# Burdens of Apprentices Caused by the COVID-19 Pandemic and How They Deal with Them: A Qualitative Study Using Content Analysis One-Year Post-Breakout

**DOI:** 10.3390/healthcare10112206

**Published:** 2022-11-03

**Authors:** Katja Haider, Elke Humer, Christoph Pieh, Paul L. Plener, Andrea Jesser

**Affiliations:** 1Department for Psychosomatic Medicine and Psychotherapy, University for Continuing Education Krems, 3500 Krems, Austria; 2Department of Child and Adolescence Psychiatry, Medical University of Vienna, 1090 Vienna, Austria; 3Department of Child and Adolescent Psychiatry and Psychotherapy, University of Ulm, 89075 Ulm, Germany

**Keywords:** apprentices, adolescents, mental health, COVID-19, burdens, resources, qualitative research

## Abstract

The COVID-19 pandemic resulted in a variety of burdens for apprentices and, as a result, in a need for resources to deal with them. The present study examined answers to two open-ended questions, which were part of a larger online survey about the burdens and resources during the pandemic from 1442 Austrian apprentices. Data collection took place from 29 March to 18 May 2021. The answers were analyzed with qualitative content analysis and revealed that most apprentices were burdened by pandemic-related restrictions and rules, the impairment of their social life, and in their mental health. In terms of resources, they mostly fell back on their social contacts, especially their friends. Besides their social life, many apprentices also used personal attitudes, leisure activities, and distractions to deal with their stresses. The study sheds light on some of the background reasons for the high psychological distress among apprentices. In order to better support young people in coping with stress, it is recommended to expand preventive and educational mental health measures targeted at apprentices, to facilitate access to low-threshold psychosocial services for young people, and improve financial support for the receipt of these.

## 1. Introduction

The COVID-19 virus outbreak over two years ago greatly impacted societies around the globe in many ways. First and foremost, lives seemed affected by the governmental restrictions to contain the spread of the virus. Since the first COVID-19 cases emerged in Austria in March 2020, the Austrian government has imposed several lockdowns, depending on current incidences. During the lockdowns, Austrians were only allowed to leave their homes to cover important basic needs, care for others in need, go to work, or undertake outdoor activities alone or with people from the same household. For young people, this also meant that their school and university education henceforth took place from home in the form of distance learning. Distance learning changed the everyday life of students, and parts of the working population in Austria were affected by a furlough (Austrian “Kurzarbeit” scheme characterized by a reduction in working hours and government-supported salaries), temporary closure of companies, and job losses [1]. For apprentices whose training combined school and work, this meant they had to face both pandemic-related changes in school and work environments.

In addition to changes in people’s day-to-day life, concerning psychological consequences of the pandemic were also becoming increasingly apparent [2,3,4]. Effects on mental health were primarily found in terms of increased stress, depression, anxiety, and insomnia [5,6,7,8]. The mental health of Austrians as a whole did not only seem to deteriorate during the first [9] and again during the subsequent lockdowns [10], but the consequences were also particularly severe for young people [10,11]. One year after the outbreak of COVID-19, 55% of Austrian school students (14–20 years old) showed clinically relevant symptoms of depression, 47% of anxiety, and 23% of sleep disorders. 37% of the students also reported having suicidal thoughts [12,13]. A female gender as well as a migration background were found to be risk factors for mental health problems in adolescents during the COVID-19 pandemic in Austria [14,15]. Similar results have been found in other countries [16,17,18,19,20,21]. Studies also found poorer mental health among students compared to the general population [22], as well as a decline in physical activity and an increase in unhealthy eating habits in adolescents [23,24,25]. This alarming trend among young people was also highlighted in several reviews [26,27,28,29,30,31,32]. 

The group of apprentices are very similar to school students not only in age, but also in terms of their mental health impacted by the pandemic: The larger investigation in which this study is embedded revealed that one year into the pandemic, 48% of Austrian apprentices showed clinically relevant symptoms of depression, 35% of anxiety, 27% of insomnia, and 50% of disordered eating [33]. Unlike school students, apprentices in Austria not only attend (vocational) school but also receive practical training while working in a company. An apprenticeship is therefore a dual vocational training that includes both theory and practice and is completed with a final apprenticeship examination after two to four years of training. The fields of work in which an apprenticeship can be completed are diverse and range from trade and crafts, industry, commerce, banking and insurance, transportation and traffic, tourism and leisure industry to information and consulting. 

The fact that apprentices are already working means that their experiences also differ to some extent from those of school students. Transitioning from a purely educational environment to the world of work represents a developmental challenge and thereby a potential risk factor for apprentices’ mental health [34]. Even before the pandemic, the mental health of apprentices was worse than those of school students [35]. To date, there are no studies on the mental health of apprentices from Austria. An Australian study showed that many apprentices had to face factors such as poor working hours, relatively small income, and insecurities regarding their jobs, which they often perceived as contributing to a general decrease in their well-being [34]. They further reported being burdened by a lack of free time, an unbalanced sleep rhythm, financial burdens, relationship problems, poor eating habits, or diminishing social support. Furthermore, the workplace was often perceived as a toxic environment, where unrealistic expectations regarding their work performance, bullying, and being exposed to aggression were on the daily agenda [34]. Another study from Australia highlighted that male apprentices working in the construction industry were at particularly high risk of committing suicide, showing poor mental health in general, and turning to alcohol and other drugs. All of these were related to workplace bullying, which many construction industry apprentices were experiencing regularly [36,37]. Additionally, the barrier to seeking help was particularly high in male-dominated sectors due to toxic masculine belief systems that imply self-reliance or the suppression of emotions [38,39]. 

A relatively large prevalence of problematic alcohol consumption during the COVID-19 pandemic was also found in Austrian apprentices working in different sectors. Especially in the age group of under 18-year-olds problematic drinking behavior increased sharply. Moreover, apprentices with clinically relevant depression, anxiety, or insomnia symptoms were more likely to show problematic drinking behavior, which again highlights the link between mental health and maladaptive alcohol consumption [40]. Compared to school students, the current sample of apprentices did not only consume significantly more alcohol but also smoked significantly more [41]. 

Since apprentices have been shown to be a burdened group even in the absence of the COVID-19 crisis [34,35,36,37,38,39] and research findings on their state of mental health [33] and consumption behavior [40,41] during this crisis added to the concern, a qualitative research approach was used to examine what apprentices felt most burdened by during the pandemic and how they dealt with these burdens. 

## 2. Materials and Methods

### 2.1. Research Design

The current study was part of a larger study investigating various aspects of mental health in Austrian apprentices during the COVID-19 pandemic. To obtain data on their mental health, an online survey was conducted from 29 March to 18 May 2021. The current study focused on two open-ended questions regarding the apprentices’ burdens (What burdens do you face as a result of the pandemic?) and resources to deal with these burdens (What helps you to cope with these burdens?). Data collection was conducted via an online survey using Research Electronic Data Capture (REDCap) [42] hosted on servers of the University of Continuing Education, Krems, Austria. Participants were recruited through several channels. Besides posts on the university website as well as on social media, e-mails were sent to the members of the Austrian Trade Union (Österreichischer Gewerkschaftsbund, ÖGB, Vienna, Austria) and the Austrian Chamber of Commerce (Wirtschaftskammer Österreich, WKO, Vienna, Austria), who distributed the study information to participants. All participants were required to agree to the data protection declaration (electronic informed consent) as well as to confirm that they were older than 14 years old. The study was approved by the data protection officer and the Ethics Committee of the University of Continuing Education, Krems (protocol code EK GZ 41/2018–2021), Austria, and was conducted according to the Declaration of Helsinki. 

### 2.2. Prevailing COVID-19 Measures

During the data collection period, a regional lockdown was reintroduced in the east of Austria (Vienna, Lower Austria, and Burgenland) on 1 April 2021, due to the newly increasing number of COVID-19 cases. Again, people living in these states had to face a 24 h curfew and leave their homes only for the coverage of basic needs, care or assistance of others, work, or outdoor activities. Schools closed again and went from classroom teaching back to distance learning. Stores that were not part of the basic supply and leisure facilities were subject to renewed closure. This meant that many employees were again affected by reduced working hours or were still on furlough anyway. Apart from these three states, all Austria had a curfew from 8 p.m. to 6 a.m. with the same exceptions for leaving the home. In addition, people were required to wear an FFP-2 mask on public transportation and in public buildings, and to keep a distance of at least two meters from other people. In addition, in Austria, it was possible to get regularly tested for the COVID-19 virus on a voluntary basis or on a mandatory basis if an infection was suspected. In Burgenland, the stricter measures were relaxed on April 19 and in Vienna and Lower Austria on 3 May 2021. Across Austria, the hotel and catering industries remained closed except in Vorarlberg [43], affecting the working hours and job security of their employees throughout the country. Reports by the Austrian Institute of Economic Research showed that young people in particular were affected by job loss [44]. This was due to several factors, such as their shorter length of employment and the shortage of temporary employment opportunities (“internships”) and apprenticeships [45]. Companies’ reluctance to hire also reduced labor market opportunities for young adults. In addition, many apprentices were still on short-time work or had to accept a reduction in working hours [44].

### 2.3. Data Analysis

Qualitative content analysis was used to evaluate the data from two open-ended questions. The principal investigator read through one-third of the answers and created a category system in the process. This involved organizing the individual categories into a thematic cluster. Once the main category system was in place, the principal investigator coded the entire data set of answers (*n* = 1442) using Atlas.ti [46] and made minor changes to the category system if a category was not yet established. Subsequently, a second coder coded 10% (*n* = 144) of the data set using the pre-existing category system to verify the consensual use of the categories. When comparing the coding of this 10% of the data set, we obtained a percentage agreement of 63.7%, Krippendorff’s c-alpha binary 0.955. Differently coded text passages were discussed by the research team and a solution was found individually for each disagreement. Most of the discrepancies occurred because the two coders dealt differently with latent and manifest content. For example, Coder 2 coded symptoms of depression described in combination, such as a lack of drive, disturbed sleep, or sadness, as depression (latent), whereas Coder 1 assigned the code ‘depression’ only when the person actually mentioned depression or being depressive (manifest; e.g., Respondent 444). The team agreed to code the manifest content only. Based on the agreements made by the team, the coding rules were further specified and then applied to the entire data set. After the changes, the percentage agreement was 96.3% inter-coder agreement using Krippendorff’s c-α-binary = 0.999. The category systems are included in Appendix A. 

## 3. Results

### 3.1. Sample

From a total of 2945 individuals, 410 were excluded since they failed to give informed consent or were under the age of 14. Another 876 did not complete the survey and were therefore also excluded. A final exclusion of 217 participants was made because they failed an attention check item. Thus, the final sample consisted of 1442 Austrians apprentices. Sample characteristics are summarized in Table 1.

### 3.2. Burdens

In the questionnaire, apprentices were asked what burdens they faced as a result of the pandemic. The results of the first open-ended question are displayed in Figure 1 and clearly show that the apprentices in Austria felt burdened mainly by the prevailing restrictions and regulations to contain the COVID-19 virus (40.64%), the impact on their social lives (36.89%) and their mental health (32.87%). 22.75% of the apprentices also mentioned burdens in education and work. 4.99% complained about physical burdens. Some apprentices faced other burdens (4.58%), did not know what they were burdened by (0.90%), or felt stressed by the fact that nothing was like it was before (0.76%). 5.62% said they were not burdened by anything as a result of the pandemic.

#### 3.2.1. Restrictions and Regulations

Within the category of ‘restrictions and regulations’ (*n* = 586, 40.64%) we saw that apprentices felt burdened especially by the requirement to wear a mask (13.18%) and restrictions on the enjoyment of their hobbies and free time (9.57%), as can be seen in Figure 2. For example, apprentices described that wearing a mask at work, especially on construction sites, was exhausting and that it often came with headaches, poorer breathing, and concentration difficulties. 9.02% of the apprentices also mentioned the curfew to be a burden. Many of them said that they did not want to be at home all the time. Closed retail and leisure opportunities were also brought up as burdens by 6.87%. Apprentices primarily suffered from not being able to go to restaurants or to parties. In addition, not being able to participate in various sports (4.85%); other COVID-19 precautions, e.g., regarding travel restrictions, distance rules, or hand sanitization (3.81%) or testing (2.15%) were also experienced as burdensome restrictions. Furthermore, apprentices also addressed the inability to attend events (1.87%), the constant changes in restrictions and regulations (0.90%), and quarantine (0.42%).

**Figure 2 healthcare-10-02206-f002:**
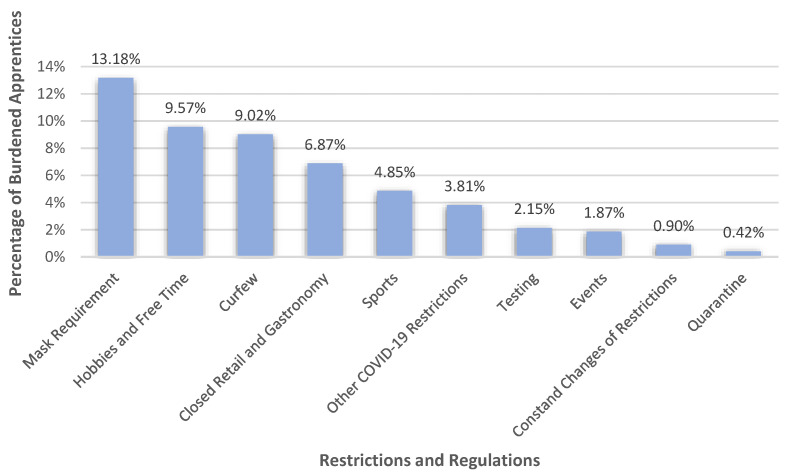
Percentages of apprentices who felt burdened by various pandemic-related restrictions and regulations.

#### 3.2.2. Social Life

Pandemic-related burdens of their social life were addressed by many apprentices (36.89%, *n* = 532). These burdens were primarily characterized by limited social contacts (33.08%). Within this subcategory of limited social contacts, apprentices mostly reported limited contact with their friends as a burden (19.42%). In comparison, limited contact with relatives or family members was only mentioned by 4.30%. Some participants stated that they were burdened by limited social contact in general (12.55%). For several apprentices, the family was perceived as a burden in the pandemic (3.33%). Some of them only reported family in general as a burden, some suffered from arguments or disagreements with the family, and some were afflicted by the spatial concentration within the family. Respondent (R) 991, for example, expressed “*Due to the pandemic, my whole family is at home most of the time. I love my family, but I need time alone and that is not possible at this time*”. Within the category of social losses (2.98%), we subsumed break-ups and impairments in the relationship, not having the possibility to make new friends, losing friends, and when participants said that their relationship with their colleagues had suffered from the current working situation.

#### 3.2.3. Mental Burdens

The third largest category addressing burdens was the category of ‘mental burdens’ (32.87%, *n* = 474), which are illustrated in Figure 3. Overall, a very broad and diverse burden spectrum could be recognized within this category. Most apprentices (13.04%) felt mental burdens that were lockdown related. Typical burdens within this subcategory were monotony, the feeling of missing out, boredom, loneliness or isolation, the feeling of being deprived of freedom, and the feeling of being locked up. With regard to monotony, apprentices, for example, reported that “*Every day feels the same (…)*” (R 27) or “*Monotonous everyday life*” (R 816). Respondent 845′s answer was an example of the feeling of being deprived of freedom: “*I am burdened by the fact that I do not feel like I have free choices. I feel oppressed and sad*”. The feeling of missing out was mostly characterized by the feeling of missing out on their youth, as, for example, described by Respondent 1134: “*Since the pandemic, I have been burdened by the fact that I can hardly go outside and thus actually miss out on my entire youth*”. The second biggest subcategory (9.71%) subsumed ‘other mental burdens’, for example, the fear of a COVID-19 infection (self-referred or of a close person), fear of the future, mental burdens of others, deaths, the news, or fear of punishments for non-compliance with the prevailing restrictions.

Moreover, apprentices mentioned some symptoms that can be associated with depression (see Figure 3) but are not unique to depression only. In addition to explicitly mentioning depression itself (4.30%), symptoms associated with depression included a lack of drive or motivation (5.06%), sleep problems (2.84%), self-doubt (0.90%), concentration difficulties (0.69%), and sadness (0.55%). Within sleep problems, apprentices, for example, revealed sleep disorders, insomnia, confused sleep rhythms, tiredness, or too much or too less sleep.

Furthermore, 4.23% of the apprentices felt stressed and 4.16% reported feeling mentally burdened in general. 1.32% of the apprentices suffered from anxiety. Lastly, a few apprentices (0.55%) reported that they suffered from either alcohol or mobile phone addictions.

**Figure 3 healthcare-10-02206-f003:**
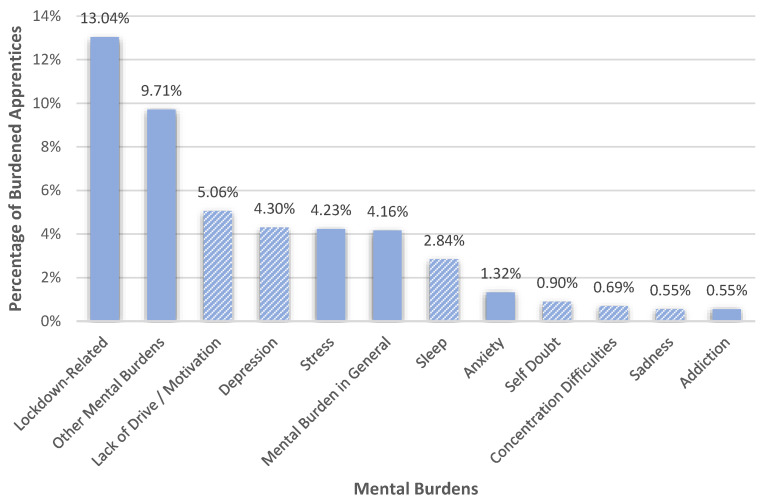
Percentages of apprentices who felt burdened in various aspects of their mental health. Columns in the striped pattern are symptoms that can also be associated with depression.

#### 3.2.4. Work and Education

The category comprising work and education-related burdens (22.75%, *n* = 328), was primarily characterized by burdens with regard to distance learning (6.24%) and home office (4.30%), as can be seen in Figure 4. Also, conditions at work (3.81%) and school (2.08%) as well as stress at school (2.70%) and work (2.29%) were perceived as burdens by the apprentices. In terms of stressful working conditions, apprentices described, for example, that as supermarket employees they were subjected to insults and threats because customers did not want to comply with the COVID-19 safety measures, e.g., mask requirement (R 1624), that working hours were poor (R 2364), customers were aggressive (R 989), that the boss was unfriendly or sometimes even angry (R 1412), or that they had to perform other activities than were normally required in their job (R 2055). The stresses related to the conditions at school referred, for example, to the fact that apprentices had to teach the learning content themselves (e.g., Rs 864, 962, 2773, 2234, 2545), that there was no group work (R 2017), that sometimes not all the learning contents were covered in class (e.g., Rs 2632, 2672), or that there was poor communication between students and teachers (R 2060). Furthermore, apprentices were worried about the final apprenticeship exam or the A-levels (Matura; 1.73%). “*(…) I am afraid that I will not pass my final apprenticeship exam because of Corona circumstances*” (R 2864), “*Thanks to Corona I have had lab 3 times so far in the whole year! 3 times! How am I ever going to pass the final apprenticeship exam???*” (R 1016) or “*I feel overloaded and overwhelmed with the Matura exam. There is hardly any time and few people to talk to. Even the teachers are overwhelmed and just try to push through their subject matter*” (R2242) were examples of worries expressed in this subcategory. The apprentices also complained about the lack of practice in their job (2.43%) and not having an apprenticeship position (1.60%) as a result of either losing their apprenticeship position or not finding one. Labor market uncertainties (0.62%), task overloads in general (0.35%), furlough (0.21%), and long journeys to the workplace (0.21%) were also reported as work- and education-related burdens apprentices had to face due to the pandemic.

**Figure 4 healthcare-10-02206-f004:**
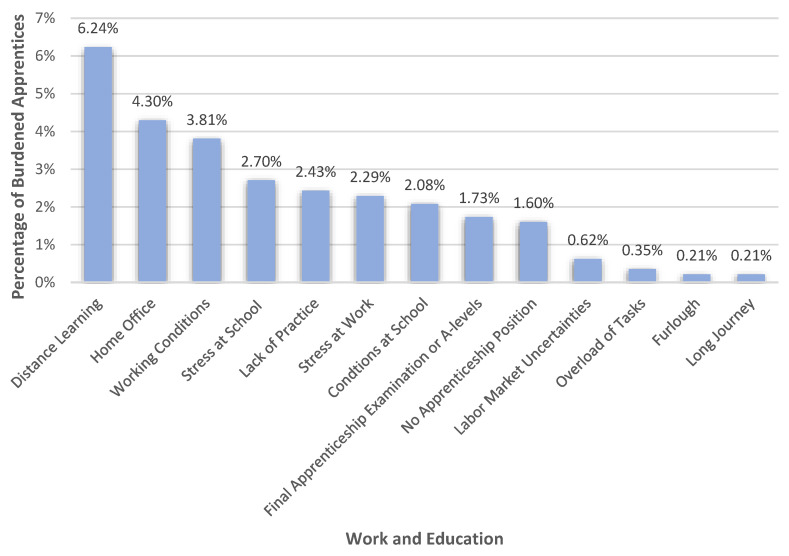
Percentages of apprentices that felt burdened in the various areas of work and education.

#### 3.2.5. Physical Burdens

Some apprentices (5.27%, *n* = 76) also reported physical burdens as a result of the pandemic. Besides impaired breathing (1.87%), headache (1.04%), postponement of medical treatments (0.28%), and circulatory complaints (0.21%), apprentices also stated various other physical complaints (2.36%), for example, nausea, skin diseases, allergies, rashes, back pain, slipped disk, or weight gain.

#### 3.2.6. Other Answers

In response to questions about the burdens they had to face as a result of the pandemic, 5.62% (*n* = 81) answered that they had to face no burdens, 0.90% (*n* = 13) said that they did not know, and 0.76% (*n* = 11) expressed their complaint about things not being like before. Furthermore, some respondents (4.58%, *n* = 66) expressed burdens in other, unspecified areas, for example, having to deal with people refusing to wear masks, household tasks, moving out, and conspiracy theories, but this category also includes financial worries or the lack of professional help, such as psychologists and psychotherapists. 

### 3.3. Resources

Following the question regarding the burdens, apprentices were asked what helped them to deal with these obstacles. Figure 5 depicts the percentages of the main resource categories. The qualitative content analysis showed that most apprentices (36.48%, *n* = 526) saw their social contacts as an important resource to deal with their burdens. 23.02% (*n* = 332) of the participants drew on personal attitudes to cope with their burdens. Furthermore, 21.71% (*n* = 313) of the apprentices perceived leisure activities as resources in times of COVID-19. Distractions (15.05%, *n* = 217) and escapism (8.67%, *n* = 125) were also used as resources. Only 2.64% (*n* = 38) noted that they consulted professional help as a resource, and 1.73% thought that their pets were helpful. 4.37% of the apprentices (*n* = 63) stated other resources. 8.18% (*n* = 118) said that nothing would help them, 2.91% (*n* = 42) explained that no help was needed, and only a few apprentices stated that they did not know what might help them (2.15%, *n* = 31).

#### 3.3.1. Social Contacts

The category ‘social contacts’, which was an important resource for 36.48% (*n* = 526) of the apprentices, consists of six subcategories. Most respondents (17.20%) stated that staying in touch and communicating with each other was an important resource. Friends could be identified as the most mentioned resource group for apprentices since 20.11% referred to them as helpful. Furthermore, 10.54% of the apprentices declared their families as a resource. Their partner (5.89%), coworkers and classmates (2.57%), as well as teachers and trainers (0.62%) were also considered as adjuvant to deal with their burdens. All social resources are shown in Figure 6.

**Figure 6 healthcare-10-02206-f006:**
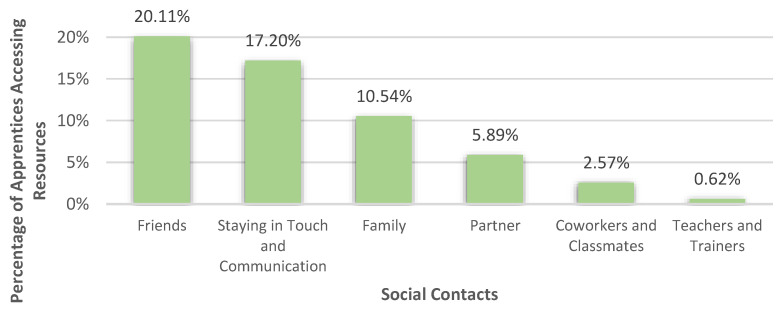
Percentages of apprentices who accessed staying in touch and communication in general and different social groups as resources to deal with their burdens.

#### 3.3.2. Personal Attitudes

23.02% (*n* = 332) of the apprentices used their personal attitude to handle their burdens. The most mentioned resource within this main category was thinking or staying positive (3.95%), as can be seen in Figure 7. 2.64% of the apprentices also stated that they had disobeyed the prevailing rules and regulations, by, for example, not wearing the mask or meeting their friends in person even though it was not allowed at that time. Some apprentices also mentioned that hope (2.64%) and taking breaks (2.36%) had helped them to cope with their burdens. Other forms of resources were relaxing (2.08%), accepting the current situation (2.01%), and behavior in compliance with the prevailing rules and regulations (1.80%), for example, staying at home, wearing masks, testing, or doing outdoor activities. Furthermore, 1.53% of the apprentices saw themselves as an important resource. They, for example, described “*Engaging with myself.*” (R 1734), “*In such a moment I sit down and talk to myself and write down what and how I have to do everything*” (R 479), or “*Taking more care of myself*” (R624). 1.39% of the apprentices mentioned suppression to deal with their burdens. Apprentices, in that matter, for example, described: “*Bottling it up and not saying anything*” (R 311) or “*I suppress these emotions hoping that things will get better*” (R 845). In addition, creating structure (1.32%) and studying (1.32%) were also referred to as helpful resources. Most of these references were made in relation to school-related problems. Few apprentices (1.25%) applied problem-solving behaviors, like opening windows, drinking lots of water, or looking for other apprenticeship positions. Few apprentices (0.55%) managed by allowing their emotions, almost all by crying. A small proportion of participants also stated motivation (0.42%) and religion (0.35%) as resources.

**Figure 7 healthcare-10-02206-f007:**
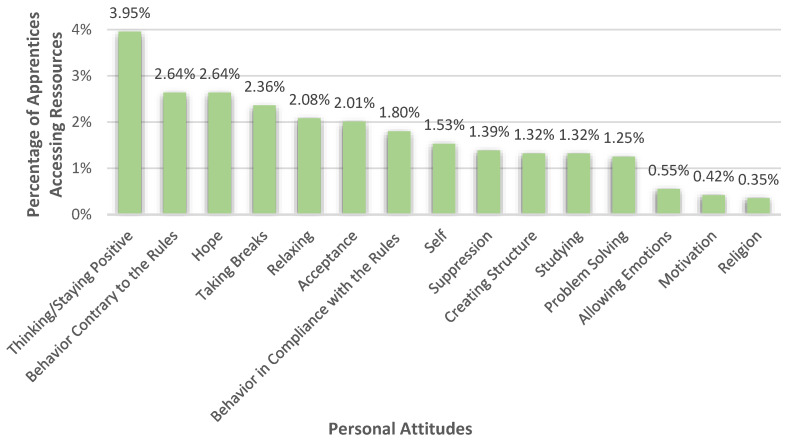
Percentages of apprentices who accessed different sorts of resources within their personal attitude.

#### 3.3.3. Leisure Activities

In the category ‘leisure activities’, which 21.71% (*n* = 313) of apprentices perceived as a resource to deal with their burdens, especially getting out into nature (8.95%) and sports (8.53%) were mentioned as helpful, which is shown in Figure 8. With regard to nature, the apprentices, for example, explained that going for walks or getting fresh air helped them to cope. In the subcategory ‘sports’ many different types of sports, like cycling, running, fitness, soccer, horse riding, dancing, and Pilates, to just name a few, and sports, in general, were stated. Some apprentices (3.68%) mentioned hobbies in general, without further specification, to be helpful. In addition, reading (1.32%), being creative (1.11%), cooking and eating (0.76%), learning something new (0.69%), and meditation (0.28%) were also stated as resources.

**Figure 8 healthcare-10-02206-f008:**
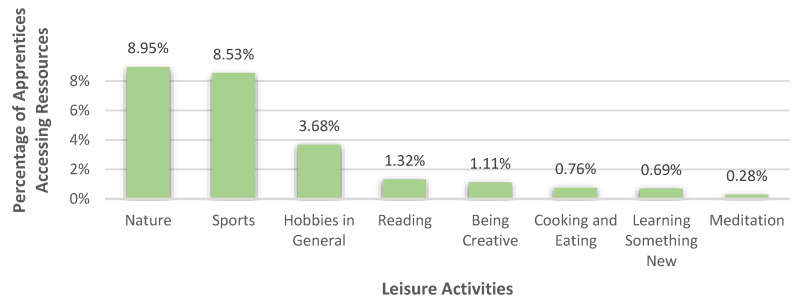
Percentages of apprentices who accessed different types of leisure activities as resources.

#### 3.3.4. Distractions

Distracting themselves was mentioned by 15.05% (*n* = 217) of apprentices as a way of dealing with their burdens. In this category, gaming and music were each viewed as helpful resources by 4.72% of apprentices. 3.47% referred to watching TV as helping them to manage and 3.05% mentioned distractions in general as resources. Also, the use of mobile phones, social media, and the internet, in general, were stated as resources by 1.87% of the apprentices. The percentages of all the accessed distractions can be seen in Figure 9. 

**Figure 9 healthcare-10-02206-f009:**
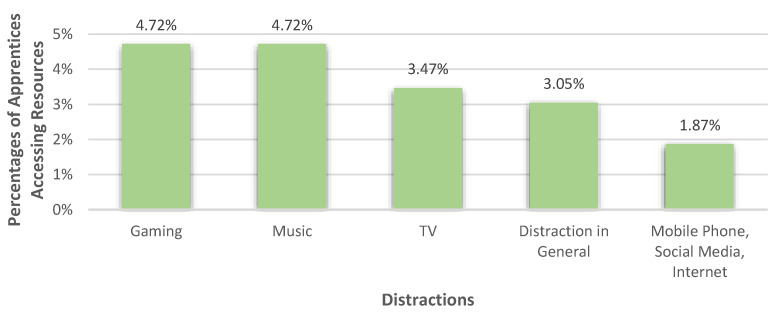
Percentages of apprentices who accessed different types of distractions as resources.

#### 3.3.5. Escapism

8.76% (*n* = 125) of the apprentices used forms of escapism to handle their burdens. 2.70% did so by sleeping and 2.22% by working. 1.04% of the apprentices turned to alcohol and 1.04% resorted to drugs. In this subcategory, the participants either mentioned cannabis or drugs in general. Furthermore, a few apprentices (0.97%) also said that cigarette smoking helped them to manage as well as escaping the situation (0.69%), unhealthy eating habits (0.55%) or self-harming (0.07%), which can be seen in Figure 10. Escaping the situation was, for example, described as escaping to a “*cold, dark, and silent room*” (R 1038). 

**Figure 10 healthcare-10-02206-f010:**
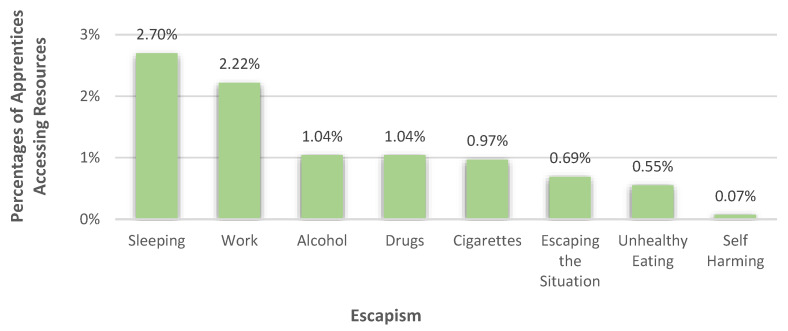
Percentages of apprentices who used different forms of escapism to deal with their burdens.

#### 3.3.6. Professional Help

Compared to all other forms of supportive behavior, only very few apprentices (2.64%, *n* = 38) sought professional help as a way to deal with their burdens. A total of 20 apprentices (1.39%) sought help from a psychotherapist; 14 (0.97%) used prescribed medications; 6 (0.42%) saw a psychologist; and 2 (0.07%) felt supported by the AMS (Public Employment Service Austria) or other labor market measures.

#### 3.3.7. Other Answers

8.18% (*n* = 118) of the apprentices could not think of anything to help them deal with their burdens and therefore answered the question with ‘nothing’. Some of the mentioned resources did not fit in any of the other categories. Thus, they were subsumed under the category ‘other’ (4.37%, *n* = 63). Examples of this diverse category were having technical devices to work with, medical attests, drinking tea, or even home schooling or the lockdowns, which usually were considered burdens by apprentices. Other apprentices (2.91%, *n* = 42) indicated that they needed no help. 2.15% (*n* = 31) did not know what might help them to deal with their burdens and 1.73% (*n* = 25) saw their pets as resources. 1.04% (*n* = 15) of the apprentices expressed wishful thinking, which was mostly characterized by the desire for the removal of certain restrictions or the end of the pandemic.

## 4. Discussion

The aim of the present study was to draw on qualitative data to provide a better understanding of the worryingly high levels of psychological distress experienced by apprentices during the pandemic, as evidenced by the quantitative results of the same study [33]. The dramatic increase across symptom groups (depression, anxiety, sleep, eating disorders, etc.), which is evident among young people worldwide [26,27,28,29,30], demands explanations. What are the underlying reasons why adolescents feel so burdened? Furthermore, in order to develop interventions and preventive measures, it is necessary to know more about the resources available to cope with stress. Although there has been much research into young people’s mental health in general, few studies have focused on apprentices. Unlike school students, they are embedded in school and work contexts, which potentially brings additional stresses, but also resources. In this study, the analysis of two open-ended survey questions enabled us to explore both the challenges Austrian apprentices had to face during the COVID-19 pandemic and how they dealt with these burdens. 

The main finding of this study was that most apprentices felt burdened by COVID-19-related restrictions and regulations. Wearing a mask all day at school and work was very often perceived as stressful, especially by those working in grocery or construction, as was not having the opportunity to find compensation after work by pursuing hobbies or leisure activities, partly due to the curfew, or going out to eat at restaurants or to do some shopping. 58.3% of the respondents—those from Vienna, Lower Austria and Burgenland—were in a lockdown situation at the time of the survey, which could explain these results to some extent. Also, a large part of the respondents worked in the fields of craft and trade, industry and commerce. In these sectors, where physical strain also plays a role, wearing masks could be more stressful than in banking, information and consulting.

Most respondents felt burdened by the limitations in face-to-face contact, and results also showed that staying in touch with friends virtually, face-to-face when permitted, or, in individual cases, even against the rules, turned out to be the most important resource helping to manage the different stresses during the pandemic. This highlights the significance of social life for apprentices. Qualitative research on the resources of similar-aged school students also showed that social contacts were reported as their main resource to deal with pandemic-related stressors [47]. In addition, peer relationships protect against psychological problems or help to cope with them better [48,49]. These findings echo well with the developmental period of adolescence, in which the significance of peer contacts increases. At the time of the survey, one-third of the respondents were in home office or on furlough or had lost their jobs due to the pandemic [33]. Given their loss of social contacts at school and at work, it seems important for young people to be able to fall back on their circle of friends as a resource.

A worrying finding of the study was the large proportion of apprentices who felt burdened by mental stress factors. There was a high diversity of psychological symptoms, such as lack of drive or motivation, tiredness, sleeping problems, self-doubt, sadness, or concentration difficulties, which might indicate depression. However, the awareness of possibly having depression seemed to be low, specifically compared to the quantitative cut-off values in the same sample. Although the quantitative cut-off analysis of the Patient Health Questionnaire (PHQ-9) [50] indicated that 48.3% of the apprentices showed clinically relevant depressive symptoms [33], only 4.30% manifestly mentioned depression in response to the open-ended burdens question. This might be because psychiatric diagnoses draw on a specific medical vocabulary that does not correspond to the language used by adolescents. However, it might also be seen as an indicator of a low mental health literacy or awareness of mental illnesses. Burns and Rapee [51] also showed that adolescents had problems labeling different signs together as depression if suicidal intent or feeling worthless were not within them. For many adolescents, the joint mention of tiredness, sleeping difficulties, changing eating habits, concentration difficulties, and losing one’s interest was often not enough to recognize depression. 

It was of further concern that despite the poor evidence on apprentices’ mental health [33], only a small proportion (2.64%) mentioned professional help, e.g., seeing a psychotherapist or psychologist, as a resource. Thus, even fewer apprentices seemed to seek professional help than the already low percentage (3.3%) of school students [47]. 

There might be different reasons for the low level of seeking professional help among apprentices and young people in general, one of which is low mental health literacy. In addition, the financial aspect might also be a barrier to seeking professional help, especially since apprentices in Austria earn only about one-third of the annual gross income of the employed in non-apprenticeship jobs [52,53] and 50% perceive their income to be too low anyway [54]. Since in Austria psychotherapeutic and psychological treatment is only partially co-financed by health insurance and fully financed treatment is very rare, the use of such services might mean a relatively high financial burden for apprentices, which many might not be able to shoulder. In spite of years of attempting to make psychosocial services more accessible [55], gaps in the psychosocial care infrastructure remain large and are becoming more evident with deteriorating mental health during the COVID-19 crisis [56]. Furthermore, the stigma that surrounds mental illnesses cannot be ignored when discussing the reasons for not seeking help from mental health professionals. For many young people, feelings of shame or self-blame keep them from seeing a psychotherapist or psychologist [57,58,59,60]. 

Nevertheless, it should be noted that low mental health literacy, poor financial status, and the stigma surrounding mental illness were factors that contributed to people not seeking professional help even before the COVID-19 pandemic. Additionally, the lockdowns made it even more difficult for people to access services from mental health professionals (face-to-face) and some psychosocial services were also simply unavailable. Apart from that, studies also showed that seeking professional help during the period of was more common in women, younger people, people with higher education [61] and those with higher levels of anxiety or depression [62].

Against this background, it seems important to facilitate lower-threshold access to psychosocial services for apprentices. In addition, it might be helpful to improve young people’s ability to label depression and other mental illnesses in order to increase their readiness for seeking psychosocial help, but also to familiarize them with the care structures so that they know where to go for help [51]. In order to further improve mental health literacy, apprentices need to be provided with well-founded and easily accessible information about mental health [63]. This will enable them to recognize psychological symptoms, assess at which point professional help for themselves or others is needed, and know where to get this help from [60,64]. We assume that, especially for apprentices, there are still few contexts in which they can obtain information about mental illnesses or psychosocial support services. In high school, there are opportunities to address these issues within the framework of certain subjects that are anchored in the curriculum (such as psychology) that do not necessarily exist at vocational schools [65,66]. Apprenticeships are organized on a company basis, and it probably depends on the employer how much importance is given to education on mental health issues. As educational workshops and mental health awareness programs have been found to lead to better awareness of mental illnesses [67] and further help destigmatize mental illnesses [67,68], we strongly propose an implementation of mental health awareness programs in (vocational) schools and workplaces. 

Some initial interventions to improve young people’s mental health have already been taken in Austria. Psychotherapeutic counseling is offered at a few secondary schools in Tyrol and Burgenland, financed mainly by the parents’ associations and schools’ directorates. As a result of positive feedback from these projects, the Austrian Federal Association for Psychotherapy (Österreichischer Berufsverband für Psychotherapie, ÖBVP) has launched ‘fit4SCHOOL’. In this project, low-threshold psychotherapeutic counseling, assistance in connecting with more extensive psychosocial support services if needed, and lectures are offered directly at schools through governmental co-financing. However, this service is only available for approximately two to four hours per week per school and must be initiated by the school itself, which suggests that there are gaps in the availability of this service in schools throughout the country [69]. Since April 2022, the government-funded initiative ‘Gesund aus der Krise’ makes it possible for children and adolescents up to 21 years of age to receive fifteen sessions of free psychological or psychotherapeutic treatment in individual or group settings [70]. Although this is a step in the right direction, the extent of psychosocial interventions for young people is still too low, and easy access is not guaranteed at the same level everywhere. Based on our knowledge of the psychosocial care infrastructure, we suspect that there are comparatively few services available in the vocational and educational contexts in which apprentices can participate. On a more general level, we would argue that expanding financial support for the receipt of psychosocial services from health insurance funds—especially for young people and those with financial constraints—would help reduce the mental burden of apprentices. This seems all the more important given that income—just like educational background—may be an important variable negatively impacting mental health [71]. Moreover, we see a need for an evenly distributed low-threshold access to psychosocial interventions at schools and workplaces across the country. It is important to include vocational schools and workplaces in order to provide an improved psychosocial care network not only for students but also for apprentices.

Besides, we still have to acknowledge that many apprentices dealt with the burdens they had to face in a healthy way. Staying in touch with their friends and family, being active, and spending time in nature, for example, might be ways to cope with the pandemic-related burdens at the present time, without needing further professional help. 

### Limitations

The findings of the present study must be considered with some limitations in mind. First of all, the data were collected within an online survey framework, where we were not able to control precisely how participants answered the open-ended questions. This explains why some answers contained several complete sentences and others consisted of only short catchwords. Although the qualitative analysis provided evidence of the challenges faced by the apprentices and what helped them deal with these burdens, it could not be ruled out that a burden or resource also applied to a person who did not mention it. Nevertheless, it is reasonable that the burdens and resources that apprentices perceived as most important were also those that they mentioned. In general, we suggest an in-depth examination of the topic, for example, by conducting qualitative in-depth interviews.

When interpreting the results, it should also be noted that no statement can be made about the intensity of a burden, or the extent of support drawn from a resource. Rather, the results show what most apprentices were stressed by and what resources they mostly fell back on. Therefore, the present study does not allow conclusions about which burdens have been perceived as the most stressful and which have been the most helpful resources. To get a better understanding of how much the apprentices are affected by certain burdens and how much certain resources help them, qualitative interviews are suited again to address these questions.

Finally, the study is cross-sectional. We do not have longitudinal data at hand, enabling us, for example, to survey changes in the experience of burdens after the end of the lockdown in Vienna, Lower Austria and Burgenland. The timing of the survey and the COVID-19 measures in place at that time could significantly affect respondents’ experiences. In addition, we cannot make any statement about how long the pandemic-related burdens have already lasted.

## 5. Conclusions

This study shows that during the COVID-19 pandemic the majority of apprentices expressed being stressed by at least one but more likely multiple different burdens, most of which regarded the pandemic-related restrictions and rules, their social life, and their mental health. Since apprentices mostly fell back on their social contacts, especially their friends, as resources, it seems reasonable to reconsider the large-scale limitation of in-person social contacts as a measure to prevent COVID-19 infections in the future. This is especially relevant for the large proportion of apprentices in distance learning, home office, on furlough, or out of work, who lost opportunities for face-to-face interaction in the contexts of work and education. We also propose the expansion of low-threshold psychosocial services for apprentices, improved financial support for the receipt of these, and educative and preventive measures in the context of vocational schools and workplaces. Young people and their needs must not be ignored, and they should have the opportunity to easily call on support. 

## Figures and Tables

**Figure 1 healthcare-10-02206-f001:**
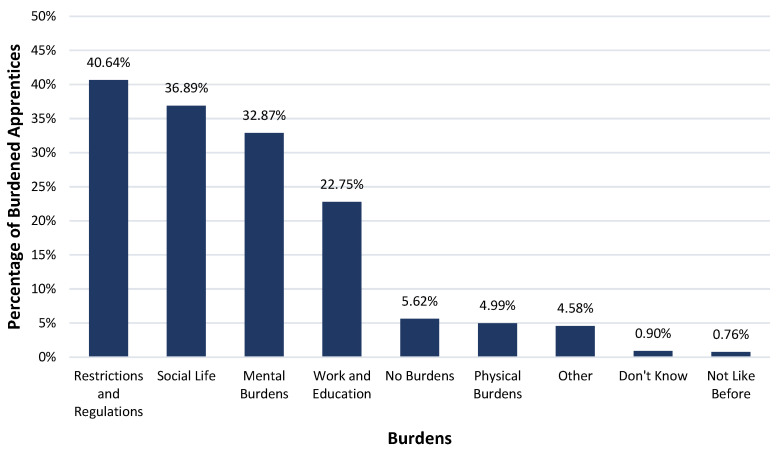
Burdens faced by apprentices as a result of the COVID-19 pandemic. The percentage of apprentices reporting one or more burdens in each of the main categories that resulted from the qualitative content analysis of the question “What burdens do you face as a result of the pandemic?”. The percentages of the burden categories may differ from the sum of the percentages in the individual burden subcategories (Figure 2, Figure 3 and Figure 4), because it may be that an apprentice was stressed by several burden subcategories (e.g., mask requirement and testing) within one category (e.g., restrictions and regulations) and thus appears in each of these subcategories but is only counted once per main category.

**Figure 5 healthcare-10-02206-f005:**
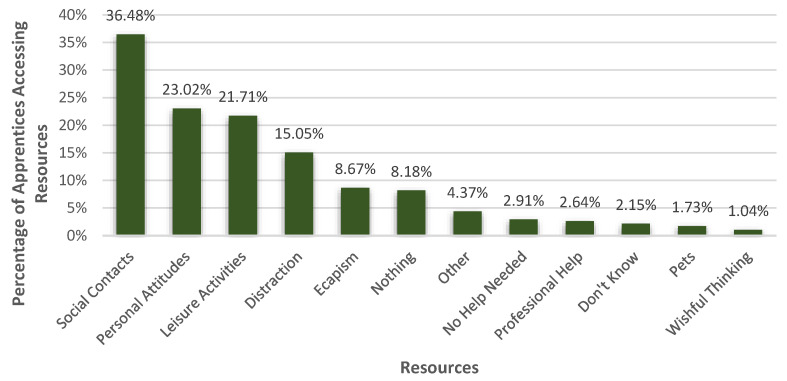
Resources apprentices accessed to deal with burdens that resulted from the COVID-19 pandemic. The percentage of apprentices accessing one or more resources in each of the categories that resulted from the qualitative content analysis of the question “What helps you to cope with these burdens?”. It is possible that the percentages of the resource categories in this figure differ from the sums of the percentages of the subcategories (Figure 6, Figure 7, Figure 8, Figure 9 and Figure 10) because some apprentices mentioned multiple resource subcategories (e.g., friends and partners) within one main category, e.g., social contacts, to be helpful.

**Table 1 healthcare-10-02206-t001:** Study sample characteristics (*n* = 1442).

Variable	N	%
Age		
15	70	4.9
16	215	14.9
17	363	25.2
18	292	20.2
19	191	13.2
20	130	9.0
>20	181	12.6
Gender		
Female	771	53.5
Male	655	45.4
Diverse	16	1.1
Migration background		
No	1022	70.9
Yes	420	29.1
Region		
Vienna	531	36.8
Lower Austria	256	17.8
Upper Austria	389	27.0
Carinthia	73	5.1
Styria	52	3.6
Burgenland	54	3.7
Salzburg	26	1.8
Tyrol	53	3.7
Vorarlberg	8	0.5
Work sector		
Craft and Trade	334	23.2
Industry	279	19.3
Commerce	223	15.5
Bank/insurance	77	5.3
Information/consulting	75	5.2
Gastronomy/tourism	66	4.6
Transport/traffic	51	3.5
Close-contact services	44	3.1
Other	293	20.3

Diverse indicates adolescents whose gender identity or gender expression does not conform to socially defined male or female gender norms. Migration background was defined as whether both parents were born abroad (second-generation immigrants) or adolescents themselves were born abroad (first-generation immigrants). Close-contact services summarizes work sectors with close body contact, such as hairdressers.

## Data Availability

The datasets generated during the current study are available from the corresponding author upon reasonable request.

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
