# Peer review of "Burdens of Apprentices Caused by the COVID-19 Pandemic and How They Deal with Them: A Qualitative Study Using Content Analysis One-Year Post-Breakout"

_healthcare, 2022, doi:10.3390/healthcare10112206_

Round 1
Reviewer 1 Report
Dear authors of the article
The topic is of general interest and the qualitative approach in the study of the impact that the COVID-19 pandemic has had on different population groups is novel, and there are few published works with the same approach. I believe that the work is relevant and pertinent for decision makers, policy makers and organizations that may have an impact on mental health. However, I would like to point out a few areas of opportunity in your work:
1- The introduction presents a lot of information concerning government measures to contain COVID-19 during different times between 2020 and 2021, however, the data collection period of your study is limited (March 29 to May 18). You might consider not including information on restrictions that are not relevant to the study period.
2- In rows 42-43 you mention "The measures were gradually relaxed and finally lifted on May 1, 2020". Would it not be appropriate to include the difference in the timing of data collection as a limitation of the study? Since you technically collected information from apprentices during the restrictive measures, once they were relaxed and finally lifted, which would affect apprentices' perception of the burden
3- In the same line of thought, in rows 120-137 that correspond to the subheading "prevailing COVID-19 measures", you mention different measures that the Austrian government took during the data collection period, however, these differ by regions of Austria (e.g. rows 121-122 "During the data collection period, a regional lockdown was reintroduced in the east of Austria (Vienna, Lower Austria, and Burgenland)...") which may be unfamiliar to the reader who does not know the country. In this regard, it is not specified in which regions of Austria the final sample of apprentices was taken, so it cannot be known whether or not those answering the questions were subject to the restrictions at the time.
4- The authors might consider including a table with more information about the sample, in which the regions of Austria to which the apprentices who answered the questions belonged could be presented.
5- There is a fundamental aspect to qualitative research, its logic indicates that the process of inference from a particular case does not lead to the generalization of the result but must derive a particular conclusion. In that sense, the authors should include more information about the sample, since the results are presented as if they represent the totality of Austrian apprentices. It is therefore important to specify to which group of apprentices the conclusions can be applied.
6- The conclusions are very specific and simply repeat the results. I recommend that the authors consider the implications for the particular group of apprentices who answered the questions.
Author Response
Dear reviewer
Thank you very much for your valuable comments on our paper and the time and effort you put into the detailed review. In the following, we would like to briefly address all points.
- We have shortened the parts of the introduction that described Covid measures prior to the start of the study in detail. Instead, we have only outlined the general consequences of the lockdown situation for young people and apprentices in particular (furlough, distance learning…). These consequences provide the context for understanding the situation of young people in Austria.
- We are unsure if we understood your second comment correctly. Did you mean to say that the survey period covered both a phase of the lockdown and its lifting, and that the changed situation could also result in a change in perceived burdens that we did not capture? Please note that (the former) line 42-43 refers to the year 2020 – the study was conducted a year later. If you recommend so, we can add this as a limitation in the final section of the paper. Before doing so, we just wanted to make sure we got the feedback right.
3.-5.Thank you for observations 3-5: the sample description was indeed very short. As you suggested, we added a table with more detailed information on our sample (e.g., distribution across regions and fields of work). First, this enables an estimate of how many apprentices were affected by the lockdown and how many were not. Second, this allows us to better contextualize the results. As you rightly pointed out, we cannot generalize but have to refer more specifically to the sample for which we are making statements. (In fact, we have conducted further analyses on the basis of our qualitative data and correlated qualitative data with quantitative parameters in a mixed methods analysis. However, these analyses would go beyond the scope of this paper, which is why we have divided the results).
- We have revised the discussion and conclusions and tried to formulate further considerations based on the results (with respect to the particular group of apprentices who answered the questions).
Please note: the uploaded document contains all the revisions made in response to reviewer 1-3.

Reviewer 2 Report
To the authors
Thank you for the opportunity to review this paper, that I find very interesting and relevant to professionals working with and researching in young people´s mental health.
The paper is clear and well written, and all main parts of the paper have the necessary information. In the discussion and conclusion, I however think that you are a bit too over-conclusive, and I think the discussion would be improved with more nuances. This is also apparent in the abstract where you conclude that ‘there is an urgent need for preventive and educational mental health measures...’. I don´t think your results so clearly demonstrate that, so it would be good to ‘loosen’ the bombastic formulations a bit.
Here are my comments:
Line 40: do you have a reference on this?
Line 49: Did not seem to deteriorate, because you only refer to two studies.
Line 72ff: Are there any studies based on Austrian apprentices? If not, it would be wise to state this, because the contexts might vary greatly between the countries.
Lines 136-137: How were the working hours and job security affected?
Line 172-173: I don´t quite understand this specific sentence – which conditions and changes did they suffer from? Maybe ‘suffer’ is a strong word, if they were asked about burdens.
Lines 434-437: Is this part of your study, or where do these results come from? If they are part of the larger survey, they would probably be better placed in the introduction.
Lines 437-438: The fact that there´s a discrepancy between the score on PHQ-9 and the open-ended responses, couldn´t that also be that the apprentices are not formally diagnosed, and do not see their burdens in line with ‘depression’? I wouldn´t say that it necessarily is connected to low mental health literacy or awareness. Psychiatric diagnosis is a very specific and medical way of labelling burdens, that I believe are not present in all young people´s way of talking about themselves and their mental burdens.
LInes 448ff: I think you need to mention one very important reason for different help-seeking behavior in general, and that´s covid itself. Because of the restrictions, it was more difficult for most people to see a professional, and many health services were unavailable during lock-downs (e.g., https://link.springer.com/article/10.1007/s11414-022-09796-2 and https://journals.plos.org/plosone/article?id=10.1371/journal.pone.0271468).
I agree, that a better access to low-threshold psychotherapeutic counseling would in many ways be helpful for young people. However, I think that you in the discussion focus a bit too much on that angle. It would be interesting to hear your thoughts on the ‘normal’, expected, responses from young people in the times of covid. In your study, it seems like a large part of the apprentices felt what is ‘normal’ to feel when you´re isolated or cut off from your daily activities, e.g., loneliness and sleep disturbances, and that most of them coped ‘in healthy ways’ with the stressors, e.g., by staying in contact with friends or being active. Those coping well, and those that only temporarily experience mental burdens, do not necessarily need professional help. And because your study is cross-sectional, you don´t get knowledge about the duration of the symptoms.
Lines 526-527: This conclusion is too bombastic, seeing that you in this study actually don´t univocally find that they have poor mental health.
Author Response
Dear reviewer,
Thank you very much for reviewing our paper and providing such valuable feedback to improve its quality. Following your recommendations, we have presented the discussion, conclusion, and abstract in a more differentiated way and have refrained from generalizing, strong conclusions. We also referred more clearly to the quantitative results of the same study, which we had in mind when formulating our conclusions. The high symptom burden across different symptom groups (depression, anxiety, sleep, eating disorders) may explain why we resorted to "bombastic" formulations. The massive increase compared to pre-pandemic data is indeed very worrying and calls for more and/or improved intervention and preventive measures. We hope that the argumentation is now a bit more balanced.
We also added a reference in line 41 (former line 40) and adapted the formulation in line 47 (former line 49).
Indeed, there are no studies on the mental health of apprentices from Austria. As recommended, we added this information in line 70f.
We also added some further information regarding job losses and the reduction of working hours (line 136-142).
You mentioned an ambiguity in Line 172f and suggested we use different wording. We have adapted the wording (line 185f). The explanation of the content of this category follows below in section 3.2.4.
Regarding your comment on lines 434-437: Yes, these results come from the quantitative part of the same study. We already reported these in the Introduction (former line 57-60). We have made it clear that this is the same study for which we report the qualitative results in the paper (line 55-59).
Your observation that apprentices cannot name their problems in terms of a psychiatric diagnosis is of course true. We have included this thought in our argumentation.
That COVID and related lockdowns were additional barriers to seeking professional help is indeed a valid point that we have now added, alongside the other factors (e.g., gender, education) you suggested in the provided papers. The additions are to be found on lines 497-504.
By noting that many apprentices found healthy mechanisms for dealing with their burdens during the pandemic, you raised an important point that we have not addressed enough in the first draft of the paper. We added a paragraph at the end of the discussion pointing out that apprentices also found healthy ways to deal with their burdens. In the limitations section, we added a statement that due to the cross-sectional nature of the study we cannot draw conclusions about the permanence of the burdens.
We have revised the conclusion and made it broader by also addressing the consequences of limited social contact. As suggested, we have taken away some of the focus from poor mental health to make the conclusion less “bombastic”.
Please note: the uploaded document contains all the revisions made in response to reviewer 1-3.

Reviewer 3 Report
The manuscript “Burdens of Apprentices Caused by the COVID-19 Pandemic and how They Deal With Them: A Qualitative Study Using Content Analysis One-Year Post-Breakout” presents a prospective study evaluating the Burdens affecting apprentices during COVID-19 in Austria.
In the manuscript, the question is original and well defined and the results provide an advance in current knowledge; the results are interpreted appropriately; all conclusions are justified and supported by the results; the article is written in an appropriate way; the data and analyses are presented appropriately; the study is correctly designed and technically sound; the analyses are performed with the highest technical standards; the methods, tools, software, and reagents are described with sufficient details to allow another researcher to reproduce the results; the conclusions are interesting for the readership of the Journal and the paper presumably will attract a wide readership; there is an overall benefit to publishing this work; the English language is appropriate and understandable.
The only comments that I would like to suggest to the authors are the following:
The introduction and discussion can be expanded and deepened with the following arguments:
(PMID: 36141222)( PMID: 35742203)( PMID: 35742047)( PMID: 35627992)( PMID: 35627970)( PMID: 35206965)( PMID: 35206922)( PMID: 35052338)( PMID: 36141296)
For the reasons listed above, my final recommendation is to accept after minor revisions the manuscript.
Best regards
Author Response
Dear reviewer,
Thank you very much for reviewing our paper and providing further references to improve both the introduction and the discussion. We have included all references you suggested, except for PMID: 35206965 and PMID: 36141296, which deal with a different subject than ours.
Please note: the uploaded file contains all the revisions adressed by reviewer 1-3.
